# Is fatigue a cue to obtain iron supplements in Odisha, India? A mixed methods investigation

Hagere Yilma  ,[1] Erica Sedlander,[1] Rajiv N Rimal,[2] Soumik Pattnaik[3]

[1]Prevention and Community Health, The George Washington University Milken Institute of Public Health, Washington, District of Columbia, USA
[2]Health, Behavior and Society, Johns Hopkins Bloomberg School of Public Health Center for Teaching and Learning, Baltimore, Maryland, USA
[3]Management, D-COR Consulting, Bhubaneswar, India

**Correspondence to**
Hagere Yilma; hyilma@gwu.edu

## ABSTRACT

**Objectives** This study aims to understand if fatigue, the main symptom of anaemia, is a health concern that cues women and their referent groups to obtain iron folic acid supplements.

**Design** This is a mixed methods study that consisted of a cross-sectional perceptual mapping and card sorting activity along with 16 focus group discussions with women of reproductive age, mothers-in-law and men. Participants of the perceptual mapping and card sorting activity were asked to compare images of anaemia-related items and concepts. Participants in the focus group discussions were asked about their daily life, aspirations and concerns among women and perceptions of anaemia and iron supplementation in the community. The quantitative data were analysed through multidimensional scaling and analyses of variance in SPSS. The qualitative data were analysed through applied thematic analysis using NVivo.

**Setting** Bhubaneswar, Odisha, India.

**Participants** Women of reproductive age (n=30), mothers-in-law (n=30) and married men (n=30) were randomly selected to participate in the perceptual mapping and card sorting exercise. A separate sample of each group was randomly selected for the focus group discussions (n=148).

**Primary and secondary outcome measures** We collected perceptions of dissimilarity between anaemia-related items, including fatigue and medical items; the extent to which these items were perceived as important to health or likable; and qualitative information about gender norms.

**Results** Cognitive maps and card sorting revealed that fatigue was conceptualised distinctly from items related to medical treatment and that perceptions around fatigue's importance to health were low. Women from the focus groups reported that fatigue is a regular part of their daily life.

**Conclusion** Our results indicate that fatigue is currently not an adequate cue to seek treatment, perhaps due to the normalisation of fatigue as a part of women's daily life.

## BACKGROUND

Anaemia is a condition characterised by a low red blood cell count that affects a quarter of all people worldwide.[1] If left untreated, anaemia can lead to several negative health outcomes, including poor mental health,[2 3] reduced work capacity,[4] pregnancy complications[5 6]

### Strengths and limitations of this study

► We adapted a marketing methodology known as perceptual mapping, coupled with a card sorting exercise and focus group discussions to understand how women of reproductive age, and their relevant referent groups (ie, mothers-in-law and men) perceive fatigue.

► Qualitative focus groups were used to minimise the ambiguity of the perceptual mapping plots.

► The size of each subsample was small (n=30 each), reducing our ability to find significant differences from the card sorting activity.

and sometimes death.[3] Women of reproductive age are the most vulnerable to anaemia,[1] as blood loss during menstruation and childbirth raises their risk.[7–10] Recent research suggests that inequitable social structures may exacerbate anaemia prevalence in women, particularly as it relates to gender norms.[1 11–14] Gender norms are a subset of social norms, which refer to expectations around how people of a specific gender should behave.[15]

The WHO has made anaemia one of six global nutritional targets.[16] Iron and folic acid supplementation is an effective strategy for the prevention and treatment of anaemia in low-income and middle-income countries.[17–19] Dosage recommendations vary by age, gender and pregnancy status, but WHO recommends regular iron supplementation for non-pregnant women of reproductive age in countries where prevalence exceeds 40%.[17]

In India, where over half of women of reproductive age are anaemic,[20] the promotion of iron supplements to children and women of reproductive age has been a part of the nutritional agenda since 1970.[21 22] However, the expansion of promotion to non-pregnant, out-of-school women only began in 2013 through the National Iron Plus Initiative.[23] Despite India's recent life-course approach to promote iron supplements to all women, consumption remains low and

anaemia prevalence is still high.[20 24] This may be due to gaps in current iron supplement distribution practices in the Indian health system. Distribution outside of clinical settings targets pregnant women or in-school adolescents; frontline workers provide tablets to pregnant women, and adolescent girls can get iron supplements weekly in school. Out-of-school, non-pregnant women do not have this access and must first go to a health facility to get tested before receiving treatment, which is typically iron supplementation for mild and moderate forms of anaemia.[25 26] This process requires an ability to detect anaemia symptoms within oneself and seek medical care.

However, the primary manifestation of anaemia, fatigue, may be ignored as a sign of illness.[27 28] In a review of the evolving conceptualisations of fatigue across time, Loriol[29] notes that, historically, fatigue is considered a necessary outcome for physical energy expenditure and, consequently, commonly attributed to a hard day's work. The result is a prevailing social expectation that the more the one expends physical energy, the more fatigued one is. This is seen in the inverse association between fatigue and socioeconomic status.[29] Therefore, women who exert a lot of physical energy in a day may not feel their fatigue is cause for alarm. Rather, they may feel that the fatigue they experience is a normal outcome of their hard work.

It is no surprise then that more than 80% of women in Odisha, India do not recognise tiredness or weakness as symptom of anaemia.[26] Studies in similar cultures have also shown that women may not readily associate fatigue with illness.[30] In fact, anaemic women who reported fatigue during pregnancy did not seek medical attention because they believed their anaemia-related fatigue was just a normal part of pregnancy.[31 32]

While the anaemia literature among non-pregnant women in India is sparse, research in other parts of the world has shown that fatigue represents a conceptually different dimension of symptoms than somatic symptoms, even when both forms share an underlying cause.[33 34] Further, Ball et al demonstrate that women's perceptions of fatigue symptoms are influenced by familial factors, whereas men hold perceptions of fatigue that are influenced by genetic factors.[33] A study in an area with high anaemia prevalence showed that culture and gender norms can shape perceptions of anaemia, which can in turn influence treatment-seeking behaviours among non-pregnant women.[13] Therefore, a similar normalisation of fatigue may exist among non-pregnant women in rural India, as expectations for labour may lead to exhaustion that is difficult to disentangle from anaemia-related fatigue. In fact, fatigue is commonly associated with activities like caretaking and employment in women but not men.[35]

In this paper, we explore the extent to which women perceive fatigue as a cue to seek medical care. While the literature around the conceptualisation of fatigue as a symptom of illness is limited, recent research in India has shown that inequitable gender norms may indirectly affect a woman's ability or desire to access iron supplements.[11–14 36] Other studies also show that women in Odisha tend to have less power within the household than their husbands, and carry the majority of the domestic workload burden.[37] As daily life is characterised by large amounts of physical labour (for both men and women), fatigue may be considered a normal part of everyday life, and thus, not a cue to obtain iron supplements. Clearly, knowledge about the prevalence and risks of anaemia is critical as risk perception may be connected to gender norms. If women believe that fatigue is just a necessary outcome of their role as women, rather than a symptom of illness, increasing risk perception may be an important first step.

Additionally, studies in India have highlighted the influence of shared-decision making within households as a key factor in anaemia prevalence among women.[38 39] To our knowledge, how this relates to different conceptualisations of fatigue by various household members—husbands, mothers-in-law and others—has not yet been explored. Yet, this is an important consideration: if women themselves do not equate fatigue with a need to seek medical care, do influential others in her home view fatigue in the same manner? Hence, our research questions are:

RQ1. Are perceptions of fatigue a cue to obtain iron supplements among women of reproductive age in Odisha, India?

RQ2. What are the differences in perceptions of fatigue across women's referent groups (their mothers-in-law and husbands)?

## METHODS

We used multiple methods to understand how conceptualisations of fatigue may relate to treatment-seeking behaviours among women of reproductive age in Odisha, India. We report findings from two methods, one quantitative and the other qualitative, used in this paper. First, we used perceptual mapping techniques and a card sorting activity to understand how our participants conceptualised fatigue.

Perceptual mapping is often used in commercial marketing[40–42] but it has recently gained momentum in health behaviour communication.[43–47] In the current study, we expand on the public health application of perceptual mapping by using this technique to understand how women understand fatigue in relation to items associated with the acquisition of iron tablets. We follow the perceptual mapping activity with a card sorting exercise to understand if cognitive (ie, perceived importance to health) and affective (ie, dislike) perceptions could inform the results of the perceptual maps.

While perceptual maps can highlight conceptual associations (or disassociations) between concepts, it is difficult to pinpoint the dimensions of perception around which these similarities or differences exist. We included a card sorting activity to probe the cognitive and affective dimensions of perception. However, a qualitative component is

also needed to delineate why these associations or disassociations may exist. Therefore, we followed the quantitative methods with qualitative focus group discussions to triangulate our findings and identify the social and cultural underpinnings of how fatigue is conceptualised.

All components of this study come from formative research conducted for the Reduction in Anemia through Normative Innovations (RANI) Project. The RANI Project is implementing and evaluating a social norms-based anaemia reduction intervention. The overall goals of the formative research components were larger than the research questions presented here; all of the goals of the formative research beyond this study are outlined the RANI project formative research protocol.[48]

### Sampling

The perceptual mapping and card sorting activity collected data from a different sample than the focus group discussion to accommodate the larger formative research goals of the methods and limit participant burden. However, all samples came from the Angul District of Odisha, India and shared the same random sampling procedure. To do so, researchers first tabulated all the homes in each of the four villages where we worked and obtained the total number of eligible participants. Based on the number of participants needed for the focus groups, against a sampling frame that consisted of the entire village, we used a proportional skip pattern that began with a randomly selected initial participant to identify households from which to select every succeeding

participant for each subgroup. From the remaining households, a similar selection process was applied to select the participants for the perceptual mapping and card sorting activity.

We conducted 16 focus groups with women of reproductive age, mothers-in-law, adolescent girls and husbands (n=124). The sample description is published elsewhere.[11] The final sample for the perceptual mapping and card sorting exercise comprised women of reproductive age (n=30), mothers-in-law (n=30) and men (n=30). All participants were required to live in the Angul District, speak Odiya, and give written consent before participating. Parental permission was obtained for all participants younger than 18 years old. A small gift of a bowl and spoon was given as incentive for participating.

### Patient and public involvement

This study did not involve patients or the public in its development or implementation. On completion of the study, results were shared with various stakeholders at a 2 days convening in Bhubaneswar, Odisha, India.

### Perceptual mapping

During the perceptual mapping exercise, we showed participants two cards at a time, each with a picture of a different anaemia-related item, and we asked them to compare how similar the two were to each other. We recorded their rating on a scale from 1 to 4, where 4 indicated the most similarity. We had 12 cards in all, and all 66 permutations (1 vs 2, 1 vs 3, … 11 vs 12) were shown

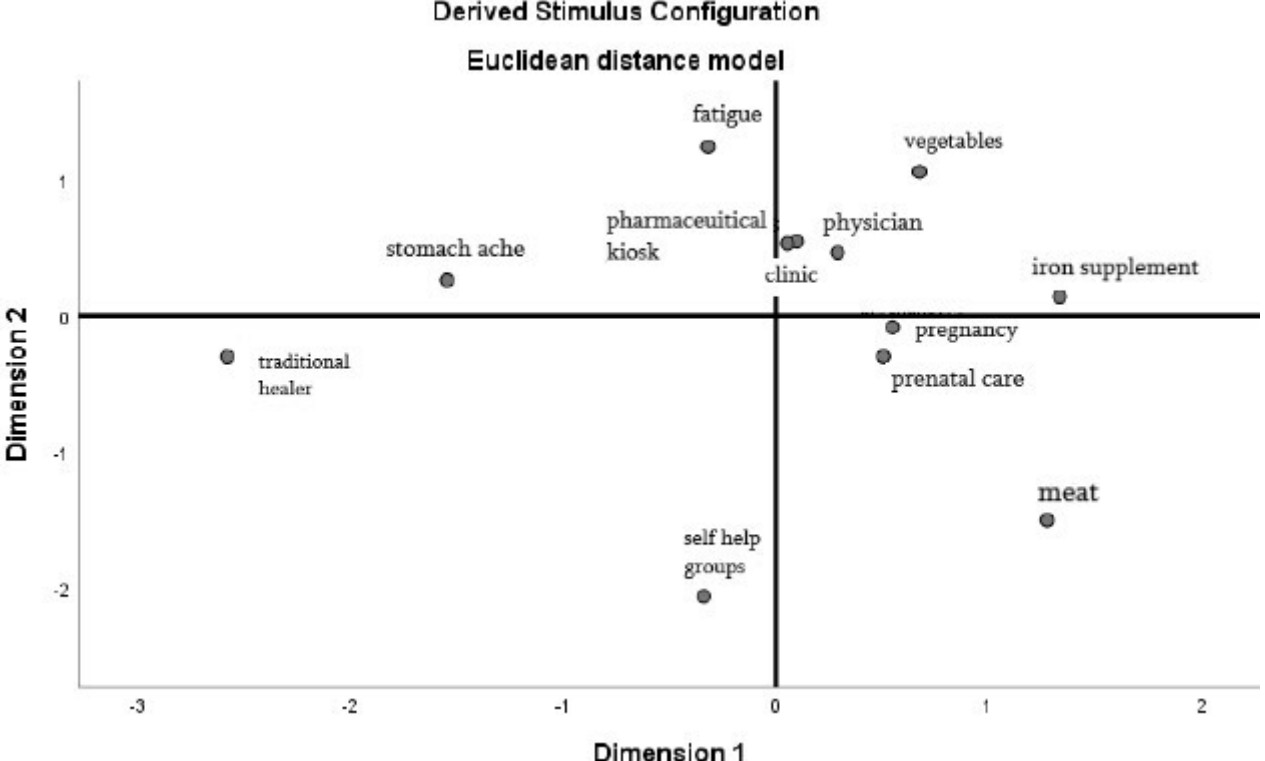

**Figure 1** Perceptual map derived for women of reproductive age. Dissimilarity plot across two-dimensions (corresponding to the x and y axes) as displayed by SPSS V.25; plot point labels have been edited for reader clarity.

to each participant. A more detailed explanation of the perceptual mapping procedure can be found in the Formative Research Protocol.[48]

### Card sorting

We also asked participants to do a card sorting exercise on completing the perceptual mapping exercise. In this portion of the study, we asked participants to place the same cards into three piles based on the items level of importance for their health and well-being. On completing this task, participants were then asked to sort the cards into another set of three piles based on the likability of the item. Data collectors recorded the compositions of each pile.

### Focus group discussions

Trained researchers (native speakers from Odisha, India) conducted all focus groups, and we matched interviewer and interviewee on gender. For each focus group, one member of the research staff observed the group while taking notes and researchers wrote field notes after each interview. Audio recordings were transcribed in Odiya and subsequently translated into English for data analysis. A more detailed description of the focus group discussion procedures can be found in a separate study.[11]

### Instruments

All instruments were developed based on discussions with local researchers and a review of the literature to ensure cultural and theoretical relevance.[25 49 50] For the perceptual

mapping and card sorting activity, we used large cards with the images of interest printed on them. The 12 cards included an image of one of the following: iron tablets, a clinic, pharmaceutical kiosks, women empowerment groups known as self-help groups, traditional healer, physician, fatigue, prenatal care, green vegetable, meat/fish, pregnancy or stomach ache. Labels of each item were also included on each card. Our colleagues in Odisha, India assessed the cards for face validity. Interview guides for the focus group discussions covered general questions about what women do on a typical day, their concerns and aspirations, and roles in the family and community. The questionnaire for the perceptual mapping and card sorting activity (online supplemental 1), along with the focus group discussion guides (online supplemental 2), is included as supplementary material.

### Analysis

#### Perceptual maps

Participant ratings were recoded to indicate dissimilarity and then organised into matrixes. We used multidimensional scaling (MDS) to graphically plot the averaged dissimilarities of the overall sample as well as each subgroup recorded (ie, women of reproductive age, mothers-in-law and men; figures 1–3). MDS is a common method to analyse perceptual mapping data.[51] Theoretically, the resulting maps can display items on multiple axes to represent underlying dimensions of perception. For simplicity and ease of interpretation, they are often

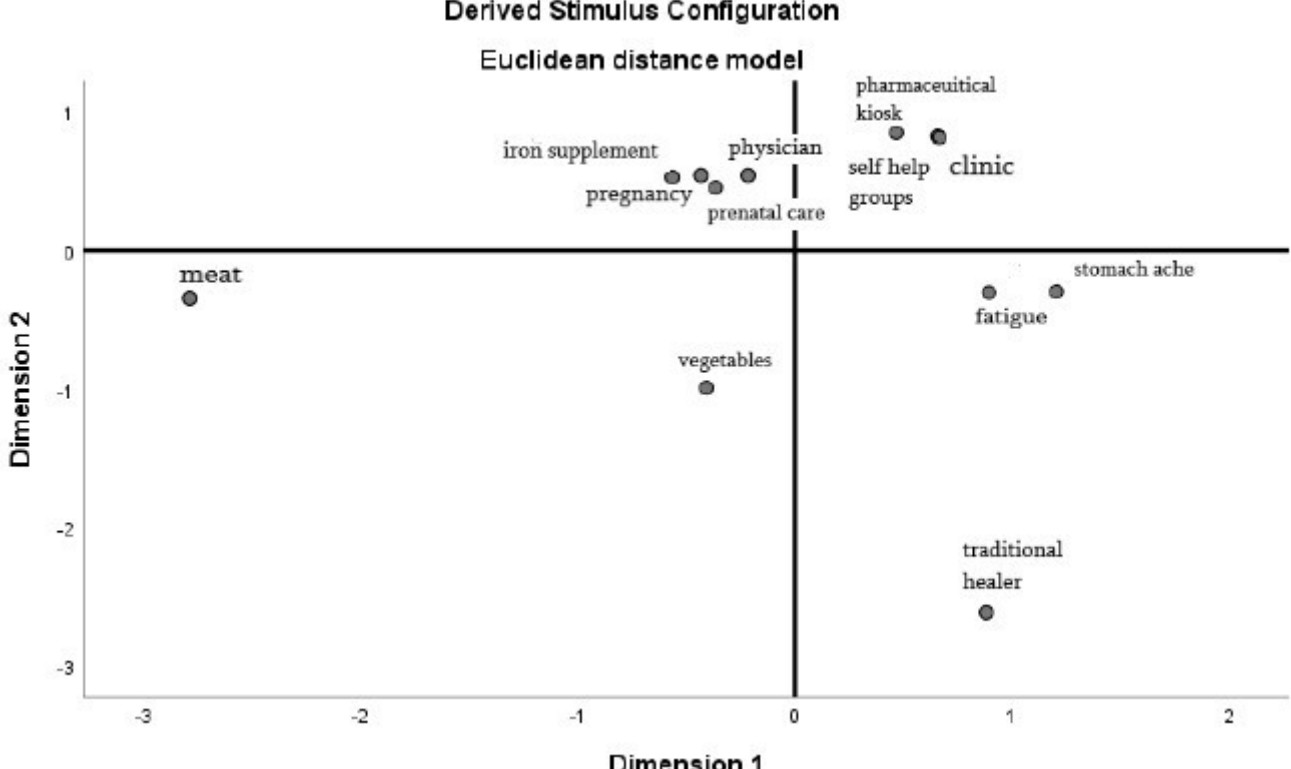

**Figure 2** Perceptual map derived for men. Dissimilarity plot across two-dimensions (corresponding to the x and y axes) as displayed by SPSS V.25; plot point labels have been edited for reader clarity.

distilled down to two (*x* and *y*) dimensions. The maps display dissimilarity so that the further two points are from each other, the more the participants perceived dissimilarity between the two items. Conversely, the closer the two points are on the map, the more the participants perceived similarity.

## Card sorting

We converted the responses from the card sorting exercise into ranks from 0 to 2, where a score of 0 was given to cards that were placed in the 'not very important to health' pile or the 'dislike' pile, and a score of 2 was given to the cards that were in the 'very important to health' pile or 'like a lot' pile, for their respective parameters. Scores for liking were reverse coded so that higher values indicated a dislike. We analysed the card sorting data using an analysis of variance across the three participant groups. We conducted post-hoc analyses to determine homogenous subsets when omnibus differences were detected. There were no missing responses. All analyses were done in SPSS V.25.

## Focus group discussions

We uploaded transcripts and the codebook into NVivo V.12 for analysis (Nvivo, 2015). We developed a draft codebook then applied thematic analysis to characterise and subsequently modify emergent themes through an iterative approach that combined data collection and analysis concurrently.[52] This allowed us to determine

when saturation occurred, when no new themes emerged from the data.

## Research team and reflexivity

Of the four authors, one is from Angul, India, one is from Nepal and two are from the USA. Their backgrounds vary from overseeing data collection efforts to social norms/gender norms, and global behavioural health. Each researcher holds a master's and/or doctoral degree and their familiarity with anaemia in India ranges from academic research to personal experience. Three of the authors, SP, ES and RR, were present for the data collection training and piloting. One of the authors, SP, was present for all data collection. None of the authors have any personal relationship to the participants or any direct benefit from examining the relationship between anaemia and fatigue.

## RESULTS
### Sample

The demographic distribution of each subgroup can be found in table 1. The average age of men in the sample was 35 years (*SD*=6.00), and that of women was 28 years (*SD*=4.76). The average age of mothers-in-law was 56 years (*SD*=7.58). The majority of women of reproductive age had completed at least secondary school, while the majority of men had not completed secondary school. The majority of mothers-in-law had not received any

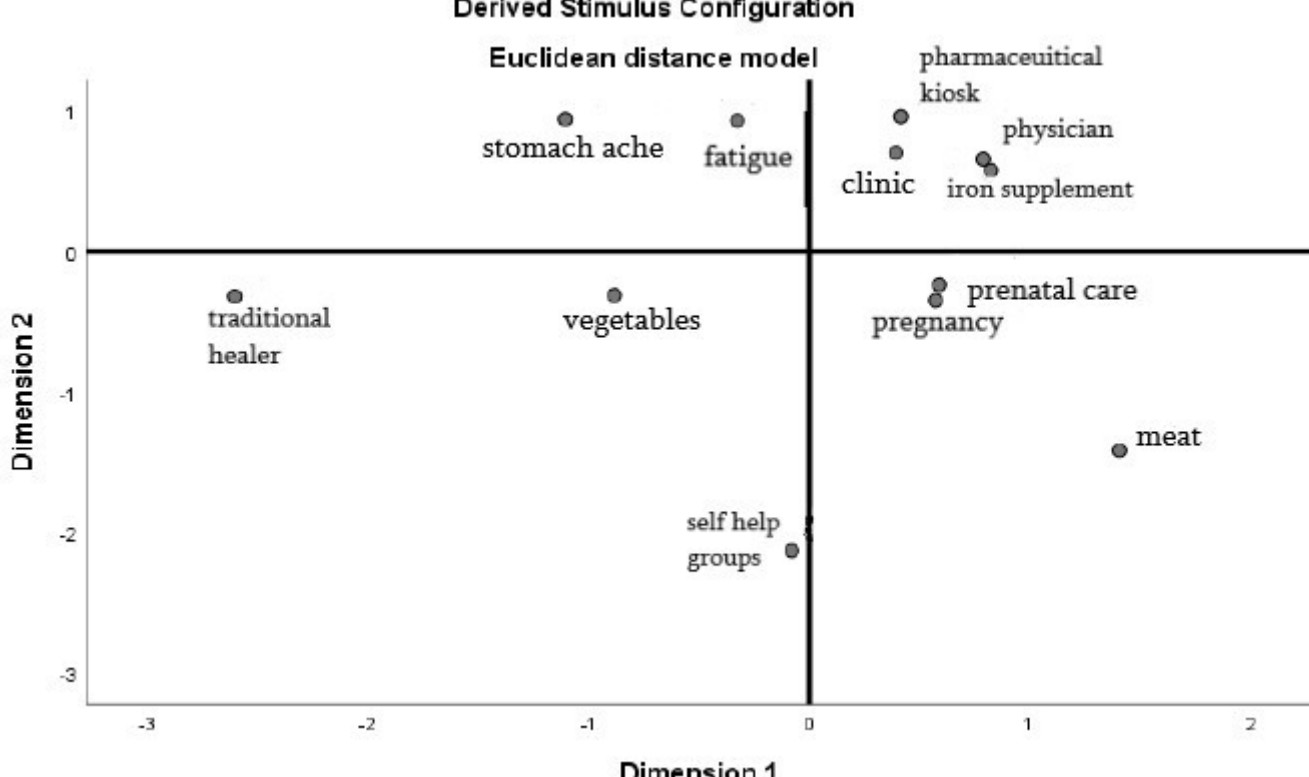

**Figure 3** Perceptual map derived for mothers-in-law. Dissimilarity plot across two-dimensions (corresponding to the x and y axes) as displayed by SPSS V.25; plot point labels have been edited for reader clarity.

**Table 1** Demographic information—perceptual mapping and card sorting

|  | Men (n=30) M (SD) | WRA (n=30) M (SD) | MILs (n=30) M (SD) |
|---|---|---|---|
| Age | 34.67 (6.00) | 27.90 (4.76) | 55.87 (7.58) |
|  | n (%) | n (%) | n (%) |
| School |  |  |  |
| None | 3 (10) | 4 (13.3) | 21 (70.0) |
| Up to primary | 12 (40) | 8 (26.7) | 6 (20.0) |
| Up to secondary | 6 (20) | 7 (23.3) | 2 (6.7) |
| Up to high secondary | 5 (16.6) | 10 (33.4) | 1 (3.3) |
| Up to tertiary | 4 (13.3) | 1 (3.3) | 0 (0) |
| Married | 30 (100) | 30 (100) | 29 (96.7)† |
| Religion—Hindu | 100 (100) | 100 (100) | 100 (100) |
| Caste |  |  |  |
| Scheduled caste/tribe | 18 (60.0) | 16 (53.3) | 17 (56.6) |
| Other backwards caste | 11 (36.7) | 12 (40.0) | 13 (43.3) |
| Other caste | 1 (3.3) | 2 (6.7) | 0 (0) |
| Children |  |  |  |
| None | 0 (0) | 3 (10) | 0 (0) |
| 1 | 4 (13.3) | 7 (23.3) | 3 (10.0) |
| 2 | 12 (40.0) | 10 (33.3) | 3 (10.0) |
| 3 | 10 (33.3) | 7 (23.3) | 11 (36.7) |
| 4 or more | 4 (13.3) | 3 (10.0) | 13 (43.3) |
| Ever taken IFA | 4 (13.3) | 26 (86.7) | 7 (23.3) |
| Currently taking IFA | 0 (0) | 3 (10.0) | 2 (9.1) |
| Diagnoses of anaemia ever | 1 (3.3) | 16 (53.3) | 4 (13.3) |
| Currently anaemic? | 5 (21.7) | 4 (14.3) | 3 (13.0) |

Under 'School', 'I' indicates incomplete and 'C' indicates complete.
*p < 0.05, **p<0.01, ***p<0.001.
†The one mother-in-law who was not married was a widow.
IFA, iron folic acid; MILs, mothers-in-law; WRA, women of reproductive age.

education. Additionally, the majority of participants had at least two children. All but one participant was married; the singular participant who was unmarried was widowed. All participants were Hindu.

While the majority of women of reproductive age had taken iron folic acid in the past (86.7%), very few were taking it at the time of the survey (10%). Only about a quarter of mothers-in-law had ever taken iron folic acid, but 9.1% were taking it at the time of the survey. None of the men surveyed reported currently taking iron folic acid, and only 13.3% had ever taken it. About half of women of reproductive age had ever received an anaemia diagnosis, and 14.3% reported currently being anaemic at the time of the survey. The demographic distribution of participants who took part in the focus group discussions was similar to the participants who took part in the perceptual mapping activity.[11]

## Perceptions of fatigue
### Perceptual maps
The perceptual maps for women of reproductive age show a conceptual separation, as evidenced by their spatial distance, between fatigue and the practical iron folic acid distribution channels for non-pregnant women; one quadrant of the map contains fatigue, while a separate one contains the items related to the iron supplement distribution channels for non-pregnant women, including physicians, clinics and pharmaceutical kiosks (figure 1). These three items are clustered together in one quadrant, indicating that they are perceived as similar to each other. Iron folic acid is also plotted in the same quadrant as this cluster, indicating that women of reproductive age may conceptualise physicians, clinics and pharmaceutical kiosks as links to iron supplements. The separation of fatigue from these iron supplement

**Table 2** Importance to health and dislike rating across participant category

| Item | Rating parameter | WRA M (SD) | Men M (SD) | MILs M (SD) | F |
|---|---|---|---|---|---|
| IFA pill | Importance | 1.67 (0.68) | 1.87 (0.35) | 1.73 (0.52) | 2.39 |
| | Dislike | 0.53 (0.73) | 0.20 (0.48) | 0.47 (0.73) | 2.15 |
| Clinic | Importance | 1.80 (0.48) | 1.93 (0.25) | 1.83 (0.38) | 0.98 |
| | Dislike | 0.23 (0.50) | 0.20 (0.41) | 0.23 (0.50) | 0.05 |
| Kiosk | Importance | 1.77 (0.50) | 1.93 (0.25) | 1.87 (0.35) | 1.44 |
| | Dislike | 0.27 (0.58) | 0.17 (0.38) | 0.13 (0.43) | 0.64 |
| Self-help groups | Importance | 1.37 (0.72) | 1.27 (0.74) | 1.57 (0.68) | 1.38 |
| | Dislike | 0.30 (0.47)[a] | 0.73 (0.74)[b] | 0.50 (0.68)[a,b] | 3.44[*] |
| Traditional healer | Importance | 0.80 (0.76) | 1.23 (0.82) | 1.03 (0.67) | 2.50 |
| | Dislike | 1.07 (0.78) | 0.97 (0.85) | 1.07 (0.69) | 0.17 |
| Physician | Importance | 1.80 (0.55) | 2.00 (0) | 1.90 (0.31) | 2.27 |
| | Dislike | 0.20 (0.41) | 0.10 (0.31) | 0.20 (0.48) | 0.61 |
| Fatigue | Importance | 0.77 (0.73)[a] | 0.80 (0.61)[a] | 0.17 (0.53)[b] | 9.66*** |
| | Dislike | 1.47 (0.68)[a] | 1.23 (0.73)[a] | 1.87 (0.35)[b] | 8.29*** |
| Prenatal care | Importance | 1.43 (0.73) | 1.40 (0.67) | 1.67 (0.61) | 1.40 |
| | Dislike | 0.70 (0.89) | 0.57 (0.50) | 0.33 (0.66) | 2.12 |
| Green veggies | Importance | 1.63 (0.62) | 1.63 (0.49) | 1.80 (0.41) | 1.06 |
| | Dislike | 0.37 (0.61) | 0.30 (0.60) | 0.13 (0.35) | 1.52 |
| Meat | Importance | 1.23 (0.68) | 0.97 (0.76) | 1.37 (0.61) | 2.62 |
| | Dislike | 0.77 (0.73) | 0.90 (0.76) | 0.60 (0.62) | 1.36 |
| Pregnancy | Importance | 1.13 (0.82) | 1.23 (0.63) | 1.57 (0.68) | 3.04 |
| | Dislike | 0.80 (0.85)[a] | 0.80 (0.48)[a] | 0.30 (0.60)[b] | 5.74** |
| Stomach-ache | Importance | 0.63 (0.67)[a] | 0.87 (0.68)[a] | 0.30 (0.60)[b] | 5.76** |
| | Dislike | 1.47 (0.68)[a] | 1.17 (0.65)[b] | 1.83 (0.46)[c] | 9.15*** |

The letters a and b denote homogenous subsets yielded from the post-hoc analysis when omnibus significance was detected.
*p < 0.05, **p<0.01, ***p<0.001.
IFA, iron folic acid; MILs, mothers-in-law; WRA, women of reproductive age.

distribution channels indicates that fatigue may not act as a cue to obtain iron supplements.

### Card sorting exercise

Confirming these findings, the card sorting exercise showed that women of reproductive age were less likely to report fatigue as important to health compared with items related to the iron supplement distribution channels for non-pregnant women (table 2). Fatigue was considered to be only somewhat important to health by women ($M$=0.77, $SD$=0.73), while the iron supplement channels, including physicians ($M$=1.80, $SD$=0.55), clinics ($M$=1.80, $SD$=0.48) and pharmaceutical kiosks ($M$=1.77, $SD$=0.50), were considered very important to health.

Differences in the conceptualisation of fatigue and the items related to iron supplement distribution channels for non-pregnant women may also be explained by differences in affective ratings (table 2). Fatigue was reported to be disliked by women ($M$=1.47, $SD$=0.68), while the other iron folic acid channel-related items were hardly disliked

at all (physician: $M$=0.20, $SD$=0.41; clinics: $M$=0.23, $SD$=0.50; pharmaceutical kiosks: $M$=0.27, $SD$=0.58).

### Focus group discussions

Women reported that fatigue, and oftentimes, extreme fatigue, was a normal part of daily life. They accepted fatigue as part of their role as wives and mothers; they take care of all of the household responsibilities including all childcare responsibilities, serve their husband and in-laws and often work outside of the home to earn extra money for the family. One woman said, "We have to work all the time, go to the mountains and fields, have children to take care of. We get no time to rest. We work in unhappiness and eat in pain" (woman of reproductive age). Women did not think of fatigue as something that they could change and they did not associate their own fatigue with the possibility of having anaemia. Although most women knew that fatigue is a symptom of anaemia, they underestimated the prevalence of anaemia in their village despite endemic rates. A frontline health worker said, "In our

village, in a year, 2-3-4 women have it [anaemia]…mainly pregnant women" (frontline health worker). This is, of course, a far cry from what the actual numbers would be, based on data from the National Family Health Survey,[20] which shows that over half of women of reproductive age are anaemic.

### Perceptions of fatigue across referent groups

### Perceptual maps

Men (figure 2) and mothers-in-law (figure 3) also perceived fatigue as different from the other items related to the iron supplement distribution channels. The perceptual map generated according to men's responses displays physicians, pregnancy and prenatal care as plotted closely together in the same quadrant. This cluster also contains iron supplements, indicating that men understand these items as existing in the same cognitive space. Similarly, the perceptual map generated for mothers-in-law displays physicians, clinics and pharmaceutical kiosks as a cluster plotted near the plot point for iron supplements. Fatigue, however, is plotted in a separate quadrant from these items related to the iron supplement distribution channels for non-pregnant women. The separation of fatigue from these channels indicates that fatigue may not act as a cue to obtain iron supplements among husbands and mothers-in-law.

### Card sorting exercise

We also explored differences in perception across participant groups through the card sorting analysis. Perceptions of importance to health didn't differ across participant groups for any of the anaemia-related items shown, with the exception of fatigue ($F=9.66$, $p<0.001$) and stomach-ache ($F=5.76$, $p<0.01$), which were significantly different between women and their referent groups. Affective perceptions (ie, dislike) related to fatigue were also statistically different across the three groups as well ($F=8.29$, $p<0.001$).

Post-hoc least significant differences tests revealed that women and men had similar perceptions of fatigue across these cognitive and affective dimensions (ie, importance to health and dislike), but mothers-in-law had significantly different perceptions. Fatigue was perceived with the same level of importance for health by both men ($M=0.80$, $SD=0.61$) and women ($M=0.80$, $SD=0.61$). However, mothers-in-law had significantly lower perceptions about the importance of fatigue ($M=0.17$, $SD=0.53$) than men ($p<0.001$) and women ($p<0.001$). Additionally, men reported a similar dislike of fatigue as women ($M=1.23$, $SD=0.73$, n.s.) while mothers-in-law disliked fatigue to a much greater extent ($M=1.87$, $SD=0.35$, $p<0.05$).

### Focus group discussions

Men acknowledged that women bear the brunt of the work at home and often face extreme fatigue but did not make the connection that anaemia could be at play. A husband from a focus group said, "They [women] don't have time to do all these works. To do the outside works

after completing the household works, they don't get enough time" (husband). Mothers-in-law felt that their daughters-in-law had it 'easier' today than when they were young and showed less empathy than men for their daughters-in-law heavy workload. They also mentioned that younger women today seek medical care more often than their generation. A mother-in-law said, "We never had those tablets [iron supplements]. We had no idea that we are pregnant until the 5th month but women of today know everything from the 1st month and run to the hospital" (mother-in-law).

## DISCUSSION

This study aimed to understand if the conceptualisation of fatigue allowed it to act as a cue for illness in women as well as referent others in their network (ie, mothers-in-law and husbands). We found that fatigue may not act as a sufficient cue for women to seek treatment for anaemia. Nor is fatigue conceptualised in a similar manner across the referent groups. These findings were notable in the perceptual maps and card sorting activity, and further manifest through the focus group discussions.

The perceptual mapping activity showed a distinction between fatigue and the cluster of items related to the iron supplement distribution channels, which included physicians, clinics and pharmaceutical kiosks. These clinical items represent necessary steps in the iron folic acid acquisition process for non-pregnant women; the disassociation between fatigue and these clinical items indicates the conceptual distinction between fatigue and a need for medical care. Additionally, the card sorting exercise showed cognitive and affective gaps in perceptions between fatigue, on one hand, and the aforementioned clinical items, on the other. In particular, women did not perceive fatigue as an important consideration for their health, although they reported dislike for it. These findings are in line with previous literature that discusses the normalisation of fatigue as an outcome of hard work rather than an illness.[29]

The qualitative results support the lack of association between fatigue and medical care in the perceptual mapping and card sorting activity. The focus group discussions clarify these findings through the identification of gender norms that may set the conditions for what mental and physical phenomenon are perceived as illness.[11–14 36] Women in the focus group discussions highlight a prevailing gender norm that women are expected to maintain a large amount of the household duties through the exertion of physical labour, which in turn, can lead to the expectation of fatigue.[29] Women in the focus groups also indicated that they prioritise the well-being of their family over their own. Thus, they accept the fatigue that they feel as a normal outcome of their household role. These unequal gender norms that place the majority of household duties on the woman may normalise fatigue in women and negatively affect their risk perception. Thus, fatigue is not an adequate cue that

illness is present, and consequently, does not act as a cue to seek treatment.

This study also aimed to investigate conceptualisations of fatigue in mothers-in-law and men. The perceptual maps created for mothers-in-law showed a slightly larger dissociation between fatigue and the clinical items related to ion supplementation than was observed in women of reproductive age. While perceptual mapping methods do not allow for the determination of statistically significant differences, the results of the focus group discussion show that the older generation of women believe that younger women have more sensitive perceptions of health. The mothers-in-law in this study harboured attitudes that were even more dismissive of fatigue than men or younger women; they did not consider fatigue important to health. These results indicate that the normalisation of symptoms, like fatigue, may be stronger among older generations of women.

We found a different conceptualisation of fatigue in men. The card sorting activity revealed a smaller gap in cognitive and affective perceptions of fatigue; while men may consider fatigue a low health concern, they also do not mind it very much. The results of the focus group discussions offer the explanation that men may not mind fatigue as much as the older generations of women because they are subject to different expectations of labour. The men in the focus group discussions show empathy for women who complete the lion's share of household work and often work outside the home. The results of the qualitative investigations coupled with the quantitative findings indicate that fatigue is normalised to a lesser extent in men than it is in women.

The conceptualisation of fatigue as illness is critical for non-pregnant, out-of-school women who are currently not given iron supplements as part of regular care by front line health workers. If women do not perceive illness when fatigue is present, they may not seek treatment for it (including asking for iron supplements) unless they are pregnant. This may exacerbate an existing gap in iron supplement distribution to non-pregnant, out-of-school women, who make up the largest proportion of women with anaemia.[20] Future anaemia reduction interventions should take into account the social and cultural underpinnings related to perceptions of illness among women.

Studies show that mothers-in-law are intimately involved in the decision-making processes among anaemic women in India.[39 53 54] If mothers-in-law continue to believe and perpetuate the idea that fatigue is not a symptom of illness, women of reproductive age may also follow suit. Even in situations where a woman does not agree with her mother-in-law, it may be difficult for her to obtain iron supplements against her mother in laws requests.[55] Therefore, in line with the recommendations of other studies, we highlight the importance of shifting mothers-in-law's perceptions of fatigue and anaemia, so that perceptions can shift in women as well.[39]

To our knowledge, this is the first study that explores the gaps in iron folic acid distribution for non-pregnant women through an examination of the social context of women living in Odisha, India. The current available health services require that non-pregnant, out-of-school, women seek treatment for their anaemia, rather than receive iron supplements unprompted. However, the most recognisable symptom of anaemia, fatigue, is not a cue for illness, and thus, does not act as a cue to seek treatment. This study shows that fatigue is not an adequate cue to seek treatment among non-pregnant women resulting in a failure to treat their anaemia. We recommend improving the available services to non-pregnant women so that they can obtain iron supplements through frontline workers. However, within the current health practice landscape of Odisha, India, we recommend increasing knowledge around the connection between fatigue and anaemia among mothers-in-laws, husbands and women to improve iron supplementation use.

## Limitations

Findings from this study provide important contributions to the anaemia-reduction literature. However, there are several limitations in this study that must be acknowledged. First, the quantitative methods used in this analysis leave gaps in interpretation. Perceptual maps are a useful tool for depicting the relative conceptualisation of items and concepts, but it gives little explanation for the dimension of perception that are depicted. We included a card sorting analysis to attempt to understand the cognitive and affective dimensions. However, there is no method to triangulate these two quantitative methods to understand if the maps are created on these two dimensions. For this reason, we have included a qualitative component, which we have triangulated with the results observed in the quantitative component. We have also used the qualitative component to provide context to the results of the quantitative assessment, despite the fact that they come from different samples. The use of different samples to triangulate may pose as a limitation, but it is worthwhile to note both components share a common sampling procedure and sampling frame.

Second, we note that there are limitations in the way that fatigue can be depicted through our quantitative methods. Participants of the perceptual mapping and card sorting activity were shown a picture of a woman experiencing fatigue, followed by a depiction of the word in the local language. It is possible that the term was too technical to be understood properly. However, the face validity of the cards was evaluated by coinvestigators in the field which minimises the chances of misinterpretation. Additionally, the majority of men and women in Odisha are literate.[56] We also note that the broader goal of the qualitative component was to understand key barriers and facilitators of iron supplementation. As a result, participants were not explicitly asked about their conceptualisations of fatigue. Hence, it is possible that their responses may shift if asked about it directly. Despite this, respondents still revealed a normalisation of hard labour among women that corresponds to normalisations of fatigue in

the literature.[29] Additionally, it is likely that not all fatigue that women are reporting is related to anaemia. Therefore, it may be useful to measure perceptions of changes in fatigue as a cue to seek medical attention, as women may have a baseline level of fatigue that is not related to anaemia at all.

**Contributors** HY, ES and RNR made substantial contributions to the conception or design of the work, the analysis and the interpretation of the work. SP also contributed to the design of the study, and led the data acquisition of data with help from ES and RNR. All authors were involved in drafting the work or revising it critically and the final approval of the version to be published. All authors are accountable for all aspects of the work.

**Funding** This study was funded by the Bill and Melinda Gates Foundation under Grant OPP1182519.

**Disclaimer** The funders had no role in study design, data collection and analysis, decision to publish, or preparation of the manuscript.

**Competing interests** None declared.

**Patient consent for publication** Not required.

**Ethics approval** This study was reviewed and approved by the George Washington University Institutional Review Board (IRB) as well as D-COR Consulting Institutional Ethics Committee (IEC). Data collectors obtained informed consent from all participants in the study.

**Provenance and peer review** Not commissioned; externally peer reviewed.

**Data availability statement** Data are available upon reasonable request. The data associated with this study can be made available upon request. Anyone who wishes to have access to the deidentified data from the perceptual mapping activity, card sorting activity, or focus group discussions can email the study data manager, Hagere Yilma (corresponding author), at hyilma@gwu.edu. All data are deidentified. Investigators are permitted to access the data for any research questions that have yet to be explored by the study team.

**ORCID iD**
Hagere Yilma http://orcid.org/0000-0002-4042-3484

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
