## [Reviewer comments · BMJ Open]

ARTICLE DETAILS

TITLE (PROVISIONAL)	Is fatigue a cue to obtain iron supplements in Odisha India? A mixed methods investigation
AUTHORS	Yilma, Hagere; Sedlander, Erica; Rimal, Rajiv; Pattnaik, Soumik

VERSION 1 – REVIEW

REVIEWER	Salla Atkins Tampere University, Finland
REVIEW RETURNED	04-Mar-2020

GENERAL COMMENTS	Dear authors, thank you for the opportunity to review this paper. I have attempted to give you detailed and concrete feedback below to allow you to improve your paper. Abstract describes the methods used poorly, more detail on e.g. analysis methods are needed Page 8 -“As daily life is characterized by large amounts of physical labor, fatigue may be considered a normal part of everyday life among women of reproductive age, and thus, not a cue to obtain iron supplements. Clearly, knowledge about the prevalence and risks of anemia is critical as risk perception may be connected to gender roles. If women believe that their role as women is to bear fatigue and not to worry about prevention efforts, increasing risk perception may be an important first step” - there is something missing in the section above. Can you clarify it? I can't get the logic of the paragraph. How does knowledge of anemia relate to gender roles in your argument? you have not shown this in the text above. You report that 20% of women know fatigue is part of anemia. Then you report above that those who know they are fatigued don't get medical attention. But do these women know fatigue is related to anemia? - Please clarify the introduction and strengthen your argument. your research questions jump straight into perceptions of fatigue, but do not establish in the area if fatigue is understood as a symptom of anemia. Your introduction is not strong enough to justify this jump. METHODS: Please explain why you choose to report the two methods used separately. Is this not a mixed methods study? It seems that you have too much data for one paper, and that focus group discussions are already in review elsewhere. Consider taking out focus group discussion information out of the paper, or justify why it is necessary. Were the participants in the perceptual mapping the same as the participants in the focus group discussions. If yes, why? If not, why not?
---

	Please justify the use of random sampling for focus group discussions (note: focus group discussions, not focus groups), as this is not a standard qualitative sampling method. Please justify why 16 focus group discussions were necessary, as this represents a large amount of data that is also not standard for qualitative studies. Please clarify your analysis, the information presented would not allow for replication. Please insert reflexivity into the results. RESULTS: Given that all the methods used concerned the same research questions (? did they?) I would expect to see increased merging of the results and analysis bringing the results together. Two paragraphs of data from 16 focus group discussions is an achievement, and makes me doubt the analysis methods used. There are also no themes presented, just descriptions of what was said, not matching the reported analysis method. Given the variation in respondents and groups, I would expect to have more nuanced analysis reporting on the differences and similarities between the different focus group discussions. Quotes should be distinguished with italics and identifiers. DISCUSSION: Please include limitations and revise discussion as you further develop the paper.
--	---

REVIEWER	Anita L. Nelson Western University of Health Sciences USA
REVIEW RETURNED	Advisory boards of AMAG and Annovera Regency 20-Mar-2020

GENERAL COMMENTS	This article is an interesting investigation of prevailing appreciation among women, their husbands and their mothers-in-law of fatigue as a medical problem which probably indicates anemia. Marketing tools were adopted appropriately to measure perceptions and importance held by those groups of fatigue. The authors posit that power of inequalities between women and their husbands is a pivotal contributor to a failure to recognize that fatigue is a significant symptom that warrants medical evaluation. Since the most common causes of anemia in this population are heavy menstrual blood losses and peripartum blood loss that are not balanced out by dietary sources of iron and folate, the authors undertook this work to investigate knowledge gaps and to identify any barriers to closing those gaps. The authors point out that in the study area only young girls and pregnant women are given iron folate supplements, but others do not. Overall, this seems to be a well conducted study that is close to being ready for publication, but there are two issues I would like the authors to consider addressing. 1. There is inconsistency in the way that the authors describe the action that should follow if a woman were to recognize that she had fatigue – in some places it is stated that she would need medical evaluation, while in many other places, it is recommended that she needed only iron supplementation. If the authors mean the latter, they should explain why a more comprehensive
---

	evaluation is not recommended, especially since no information was ascertained about her menstrual blood loss or diet. 2. The second issue is more subtle. There is undercurrent in the work that the men are dismissive of the woman's fatigue (anemia). Do we have any data on anemia among men in the study population? The mothers-in-law are similarly portrayed as being unhelpful because they perceive that they had it even worse; are there data to suggest they are wrong? I think we can be more sympathetic of their plight. Are gender norms assumed to be cultural norms about gender roles? I believe that when gender roles are separated for culture, the point of the study gets blurred How was fatigue defined? Were examples given when subjects were asked about it? Is it possible that the subjects were asked such a technical term? Page 2, line 14 – Males is an adjective you need the noun – men. Page 4 Is it really “innovative” or did you adopt these approaches from another discipline (marketing)? Page 6, line 19 – Missing the word women after pregnancy. Page 7, line 34 – When you suggest that knowledge about risks of anemia is connected to gender roles, how do you rule out general ignorance as a cause of that knowledge gap? Page 7, line 36 – Why mention “prevention efforts” here. Everywhere else the focus has been on therapy with iron. Is this a new frontier – do you want to suggest that asymptomatic women receive iron too? The rest of the work has focused on symptomatic women. This shift blurs the argument. Page 9, line 28 – No need for abbreviations (WRA, MIL). If you do need to use those abbreviations, please define them the first time you use them. Page 10, line 45 – Really could use some references here when you claim to have reviewed the literature. Curious again that HMB was never investigated in this way. Page 11, line 33 – Why was it useful for higher scores to indicate a dislike? Did it help the display? Page 12, line 12+ - What were your eligibility criteria? Why did women agree to participate? Were the subjects given any incentives? Did their husbands have to approve of their participation? If so, did that introduce any biases? Page 18, line 1 – Why do the women think iron is important when they are pregnant? Page 18, lines 35-43 – Something appears to be missing. Page 19, lines 10-12 – Difficult to demonstrate when both genders agree to the roles and have such similar results in your evaluations. Can you discuss more clearly? I would be happy to review any revisions the authors may decide to submit.
--	--

VERSION 1 – AUTHOR RESPONSE

Responses to Comments Provided by Reviewer 1 (reviewer comments are presented in bullets):

- Dear authors, thank you for the opportunity to review this paper. I have attempted to give you detailed and concrete feedback below to allow you to improve your paper.

Thank you for your thorough review of our manuscript. We have responded to each of your comments to the best of our ability and look forward to your second read of our study.

- Abstract describes the methods used poorly, more detail on e.g. analysis methods are needed

We made the decision to write a sparse methods section for the abstract to stay in line with the journal guidelines. However, in light of your comment, we have made additions to this section of the abstract.

- Page 8 -“As daily life is characterized by large amounts of physical labor, fatigue may be considered a normal part of everyday life among women of reproductive age, and thus, not a cue to obtain iron supplements. Clearly, knowledge about the prevalence and risks of anemia is critical, as risk perception may be connected to gender roles. If women believe that their role as women is to bear fatigue and not to worry about prevention efforts, increasing risk perception may be an important first step” there is something missing in the section above. Can you clarify it? I can't get the logic of the paragraph. How does knowledge of anemia relate to gender roles in your argument? you have not shown this in the text above. You report that 20% of women know fatigue is part of anemia. Then you report above that those who know they are fatigued don't get medical attention. But do these women know fatigue is related to anemia?

This is an important point. We now realize the previously used language is confusing—the issue is not that women do not know they are fatigued. Rather, the issue is that women do not attribute their fatigue to illness. We have made substantial changes in the introduction section to make this point clearer. We have also included more literature as evidence for the normalization of fatigue as an outcome of hard work rather than illness, and the gender norms related to labor expectations of women as compared to men.

- Please clarify the introduction and strengthen your argument.

We have made substantial revisions to this section. We hope the new introduction is clearer and makes our arguments more rigorously.

- Your research questions jump straight into perceptions of fatigue, but do not establish in the area if fatigue is understood as a symptom of anemia. Your introduction is not strong enough to justify this jump.

In light of this suggestion, and the previous suggestions relating to the introduction section, we have made substantial additions to this section (Introduction, p.6-7). These additions include a discussion of the literature around perceptions of fatigue and the role that gender norms can play in influencing health perceptions. We are confident that these changes strengthen our argument considerably.

- Please explain why you choose to report the two methods used separately. Is this not a mixed methods study?

We agree and have made these changes so that the methods are reported together (Methods, p.9-14).

- It seems that you have too much data for one paper, and that focus group discussions are already in review elsewhere. Consider taking out focus group discussion information out of the paper, or justify why it is necessary.

We gave this issue considerable thought, it is important to note that the methods presented here come from the formative research of the Reduction in Anemia through Normative Innovation (RANI) project. Thus, the goals of the methods were broader than what is presented here (the broader

formative research methods are summarized in a published protocol¹; the qualitative methods aimed to understand *all* of the barriers and facilitators to iron folic acid supplementation in women of reproductive age in Odisha. The results of the inquiry on the broader barriers and facilitators are published in another article.² However, a narrower analysis of the qualitative findings overlaps with the more specific goals of the perceptual mapping and card sorting activity. A limitation of the perceptual mapping method is an inability to understand the dimensions of perception onto which key concepts are mapped. The inclusion of findings from the focus group discussions were necessary for not only triangulating the results, but also for unpacking the social and cultural underpinnings of the disassociation we found between fatigue and medical items. We have now included text that justifies the use of both quantitative and qualitative data in the manuscript (Methods, p. 9-10).

- Were the participants in the perceptual mapping the same as the participants in the focus group discussions? If yes, why? If not, why not?

This is an important point that we regret was not clear in the original manuscript. We maintained separate samples for the quantitative and qualitative components for two reasons. First, we did not want participants' responses in one research to influence what they told us in the other. Second, we deemed it necessary to minimize participant burden. We have edited our manuscript to make this clearer (Methods, Sampling, p. 10). However, we acknowledge that there are limitations to this decision and have listed them in our discussion section (Discussion, p.23-24).

- Please justify the use of random sampling for focus group discussions (note: focus group discussions, not focus groups), as this is not a standard qualitative sampling method.

Departing from more traditional purposive sampling techniques, we chose to randomly sample participants for the focus groups for two reasons. First, we did not want to rely on choosing participants identified by leaders in the community, for fear that such a process would result in the selection of participants who are not representative of the larger community. They would likely be more articulate and knowledgeable than their peers (because leaders could be biased to represent their community in a positive light). They would likely also be more experienced in research (having been selected in other projects like ours in the past). Second, because anemia is so highly prevalent (range of 70% in our communities) and because it is not a stigmatizing condition (like, say HIV/AIDS), we did not fear that strong privacy or confidentiality issues would prevail, which would negate having people unknown to each other participate in the same focus group). Thus, we chose to have participants with a diverse background to obtain diverse perspectives, which was achievable through a random selection process.

- Please justify why 16 focus group discussions were necessary, as this represents a large amount of data that is also not standard for qualitative studies.

We have noted above that the qualitative methods had a broader goal of identifying all of the barriers and facilitators for iron folic acid consumption. Thus, the focus group discussion comprised 16 focus groups for this larger purpose, and we extracted data from that larger corpus when it was relevant to this particular research.

¹ Authors (2018; masked for blind review). The RANI Project: A socio-normative intervention to reduce anemia in Odisha, India: A formative research protocol. Gates open research, 2.

² Authors (2020). Moving beyond individual barriers and identifying multi-level strategies to reduce anemia in Odisha India. BMC Public Health, 20, 1-16.

- Please clarify your analysis, the information presented would not allow for replication.

In terms of the qualitative research, the small sample size limits generalizability to urban areas or other areas outside of India. We've also added this point in our limitations section. However, random sampling may provide a more representative sample and therefore make the findings more generalizable than a purposive sample. Additionally, a larger analysis of the qualitative data was conducted to identify all of the barriers and facilitators to iron supplement use. A more detailed description of the analysis procedures are described in a broader qualitative study written by the RANI project³.

- Please insert reflexivity into the results.

We've now added a statement on reflexivity (Methods, p. 14).

- Given that all the methods used concerned the same research questions (? did they?) I would expect to see increased merging of the results and analysis bringing the results together.

We agree that results can lend themselves to being merged. However, the purpose of including the qualitative findings was to do more than triangulate our findings from the quantitative. The qualitative methods were also meant to unpack the perceptual mapping findings and identify the cultural or social underpinnings. Thus, we found that the results are better presented with the qualitative findings following the quantitative.

- Two paragraphs of data from 16 focus group discussions is an achievement, and makes me doubt the analysis methods used. There are also no themes presented, just descriptions of what was said, not matching the reported analysis method. Given the variation in respondents and groups, I would expect to have more nuanced analysis reporting on the differences and similarities between the different focus group discussions.

We agree that 16 focus groups yields much richer data than as presented here. However, as noted, the broader goal of the focus group discussion were to understand the overall barriers and facilitators for the RANI project; the paper written here takes advantage of the overlap between the broader goals of the focus group discussions and the more specific goals of the perceptual mapping activity. Thus, while this is a mixed methods study, the qualitative findings presented here are sparser than is presented in the report of the overall findings from the RANI project qualitative research as they are meant to only triangulate and unpack the findings of the quantitative portion in this paper. Put another way, while we totally agree that the more traditional way of analyzing qualitative data would, indeed, be to look for underlying themes (and our paper on facilitators and barriers does exactly that), our purpose for making use of the qualitative data in this paper was somewhat different: we were interested in seeing how the qualitative data could inform and contextualize the dimensions onto which our perceptual mapping data's findings could be mapped.

- Quotes should be distinguished with italics and identifiers.

We have now italicized all quotes and added identifiers in parentheses. Please note that we are unable to add age at the request of the Institutional Review Board (IRB).

³ Authors (2020). Moving beyond individual barriers and identifying multi-level strategies to reduce anemia in Odisha India. BMC Public Health, 20, 1-16.

- Please include limitations and revise discussion as you further develop the paper.

Thank you for this note; we have now updated our discussion section accordingly (Discussion, p. 23-24)

Responses to Comments Provided by Reviewer 2 (reviewer comments are presented in bullets):

- This article is an interesting investigation of prevailing appreciation among women, their husbands and their mothers-in-law of fatigue as a medical problem which probably indicates anemia. Marketing tools were adopted appropriately to measure perceptions and importance held by those groups of fatigue.

Thank you for your careful review of our manuscript. We sincerely appreciate your time and consideration and are glad to hear that you find our methods suitable for this study.

- The authors posit that power of inequalities between women and their husbands is a pivotal contributor to a failure to recognize that fatigue is a significant symptom that warrants medical evaluation. Since the most common causes of anemia in this population are heavy menstrual blood losses and peripartum blood loss that are not balanced out by dietary sources of iron and folate, the authors undertook this work to investigate knowledge gaps and to identify any barriers to closing those gaps. The authors point out that in the study area only young girls and pregnant women are given iron folate supplements, but others do not.

We believe that our manuscript offers great contributions to the anemia-reduction literature for reasons that you have so kindly articulated.

- Overall, this seems to be a well conducted study that is close to being ready for publication, but there are two issues I would like the authors to consider addressing.
 - There is inconsistency in the way that the authors describe the action that should follow if a woman were to recognize that she had fatigue – in some places it is stated that she would need medical evaluation, while in many other places, it is recommended that she needed only iron supplementation. If the authors mean the latter, they should explain why a more comprehensive evaluation is not recommended, especially since no information was ascertained about her menstrual blood loss or diet.

Thank you for pointing this out. We believe that a clearer explanation of the health systems can clarify this point. Currently, iron folic acid is promoted for the treatment of mild and moderate anemia. However, the process of obtaining iron folic acid for non-pregnant out of school girls requires medical evaluation, as they are not handed iron folic acid by frontline workers. More serious medical intervention is typically only required of only severe anemia, which is much rarer and presents more salient symptoms. Thus, when we mention medical treatment of anemia, we are primarily referring to iron supplementation. We have made changes to the introduction section to make this clearer (Introduction, p. 6, paragraph 1)

- The second issue is more subtle. There is undercurrent in the work that the men are dismissive of the woman's fatigue (anemia). Do we have any data on anemia among men in the study population? The mothers-in-law are similarly portrayed as being unhelpful because they perceive that they had it even worse; are there data to suggest they are wrong? I think we can be more sympathetic of their plight.

In light of this comment, we have rewritten the narrative on findings around the perceptions of men (Discussion, p. 21, paragraph 3). We think that this rewrite makes it clear that our findings do not indicate that men are dismissive of women's anemia. In fact, they are sympathetic to the burden of labor endured by women. The daily demands of men do not result in a normalization of fatigue to the same extent observed in older generations of women. National Family Health Survey data in India⁴ indicate that anemia prevalence is relatively low in men (compared to women), which may be linked to their more accurate perceptions of fatigue. In India, half of women of reproductive age, compared to 23 percent of men, have iron deficiency anemia.

Additionally, while we understand your concerns about the portrayal of mothers-in-law, we do not believe that our narrative is not sympathetic to their plight. It is most likely the case that the greater normalization of fatigue in older generations of women observed in this study is rooted in greater hardships than are endured in younger generations. However, despite the very real hardships that their perceptions may be rooted in, mothers-in-law can negatively influence the health seeking behaviors of their daughters-in-law if they don't perceive fatigue to be a symptom of illness. Thus, we find it important to highlight the need for shifting the perceptions of mothers-in-law as well.

- Are gender norms assumed to be cultural norms about gender roles? I believe that when gender roles are separated for culture, the point of the study gets blurred

We appreciate the opportunity to discuss this. As we mention in the paper, gender norms are a subset of social norms. The extent to which one ascribes to gender roles is an indicator of gender norms. For example, in areas with more rigid gender norms, it may be a shock to see a woman as a pilot. However, in an area where gender norms are less rigid, gender roles may be fluid and seeing a woman in this type of job would not cause a second look. In the context of Odisha, gender roles are rather strictly ascribed to, indicating strict gender norms. We realize that the language may be confusing and takes away from our larger point. Given that we are focused on gender norms (as gender roles are more strictly ascribed to), we have replaced all mentions of gender roles with gender norms.

- How was fatigue defined? Were examples given when subjects were asked about it? Is it possible that the subjects were asked such a technical term?

Fatigue was shown as an image to participants (showing a woman experiencing tiredness), which is typical in perceptual mapping exercises. Data collectors also said the word 'fatigue' in the local language along with showing the card. We also conducted extensive pilot testing to ensure that participants understood the exercise and the images that we were using to convey different words. We have made this clearer in our methods, but in case that it is not, we refer readers to the more detailed formative research protocol of the RANI project⁵ (the project this study stems from). Given that there was a visual representation of fatigue shown to the participants selected alongside coinvestigators from the field, we believe that there were not issues in interpretation of the term. However, we now include this as a potential limitation in our discussion (Discussion, p. 23-24).

- Page 2, line 14 – Males is an adjective you need the noun – men.

We have made sure that we now refer to our male participants as "men" where appropriate.

- Page 4 Is it really "innovative" or did you adopt these approaches from another discipline (marketing?)?

⁴ International Institute for Population Sciences. (2017). *National family health survey (NFHS-4), 2015-16*. India. Mumbai: IIPS.

⁵ Authors (2018; masked for blind review). The RANI Project: A socio-normative intervention to reduce anemia in Odisha, India: A formative research protocol. *Gates open research*, 2.

We have now adapted our wording to encompass more accurate language. The sentence now reads “We adapted a marketing methodology known as perceptual mapping...”

- Page 6, line 19 – Missing the word women after pregnancy.

Corrected.

- Page 7, line 34 – When you suggest that knowledge about risks of anemia is connected to gender roles, how do you rule out general ignorance as a cause of that knowledge gap?

While we do not think we can rule out general ignorance as a cause for the gap in conceptualizing fatigue as a symptom of illness, we do think that gender roles, or rather gender norms, play a role beyond general ignorance. We refer to a study conducted in Sri Lanka to support this point⁶. This study found that for men, perceptions about fatigue are influenced by genetic factors more than familial environments. Whereas, among women, perceptions of fatigue are related to normative influences more than biological factors. We also know that fatigue is generally considered a necessary outcome of hard work. Therefore, in areas where gender norms dictate large amounts of labor for women, women are more likely to consider their fatigue as an outcome of their work, rather than symptom of illness. We have made substantial edits to make this point clearer in this version of the manuscript.

- Page 7, line 36 – Why mention “prevention efforts” here. Everywhere else the focus has been on therapy with iron. Is this a new frontier – do you want to suggest that asymptomatic women receive iron too? The rest of the work has focused on symptomatic women. This shift blurs the argument.

We have restructured this sentence in a manner that does not mention prevention efforts (Introduction, p.8). The rest of the paper continues to focus on treatment seeking as the primary behavior.

- Page 9, line 28 – No need for abbreviations (WRA, MIL). If you do need to use those abbreviations, please define them the first time you use them.

We have made the necessary changes throughout the manuscript so that abbreviations are no longer used.

- Page 10, line 45 – Really could use some references here when you claim to have reviewed the literature. Curious again that HMB was never investigated in this way.

We have now included three of the references we used the most in instrument development (Methods, Instruments, p.13).

- Page 11, line 33 – Why was it useful for higher scores to indicate a dislike? Did it help the display?

This was done mainly to facilitate the interpretation of results, which became more intuitive if higher scores indicated dislike as fatigue was our primary item of focus; it is more intuitive to think of a person as disliking fatigue than liking it.

- Page 12, line 12+ - What were your eligibility criteria? Why did women agree to participate? Were the subjects given any incentives? Did their husbands have to approve of their participation? If so, did that introduce any biases?

⁶ Ball, H. A., Siribaddana, S. H., Sumathipala, A., Kovas, Y., Glozier, N., Rijdsdijk, F., ... & Hotopf, M. (2011). Genetic and environmental contributions to the overlap between psychological, fatigue and somatic symptoms: a twin study in Sri Lanka. *Twin Research and Human Genetics*, 14(1), 53-63.

We have now outlined our eligibility criteria and incentives more clearly in the methods section (Methods, Sampling, p 10-11). We'd also like to note here that women did not have to obtain formal approval from their husbands to participate, so we did not see any biases from this.

- Page 18, line 1 – Why do the women think iron is important when they are pregnant?

We'd like to turn to a point made in our introduction paragraph to answer this (Introduction, p.6, paragraph 1). Pregnant women are given iron supplements outside of the health clinic as frontline workers distribute them to their homes regularly for the duration of their second and third trimester. For this reason, norms around iron taking behavior in pregnant women are much greater than in non-pregnant women, so there is generally a greater perceived importance for iron supplements when pregnant. However, the current health system puts iron supplements in the hands of pregnant women, regardless of if they ask for it. Therefore, beyond just general perceived importance, pregnant women have the advantage of being ambivalent towards their fatigue, while non pregnant women must recognize their fatigue as a symptom of illness and seek treatment,

- Page 18, lines 35-43 – Something appears to be missing.

We have restructured our discussion section and you will see that we have addressed all gaps in this section.

- Page 19, lines 10-12 – Difficult to demonstrate when both genders agree to the roles and have such similar results in your evaluations. Can you discuss more clearly?

We have restructured our discussion section to bring gender norms (rather than roles) to the forefront and provide a better argument for why unequal gender norms can result in differing perceptions of fatigue. We believe that our discussion section is now outlined in a way that makes these points clearer.

Thank you again to the editors at BMJ Open and the reviewers assigned to this paper. We cannot stress our appreciation for your time and careful consideration. We are eager for your second review of our manuscript and await your response.

Sincerely,

Hagere Yilma, MPH (corresponding author); Erica Sedlander, DrPH; Rajiv N. Rimal, PhD; Soumik Pattnaik, MS

VERSION 2 – REVIEW

REVIEWER	Anita L. Nelson Western University of Health Sciences USA
REVIEW RETURNED	09-Jun-2020
GENERAL COMMENTS	BMJOPEN-2020-037471R1 Minor Revisions Is fatigue a cue to obtain iron supplements in Odisha India? A mixed methods investigation This manuscript is definitely improved, although it could benefit from tight editing to reduce its length. Much of the confusion has been eliminated about recommendations for women to follow should they make the association between fatigue and anemia. A

	fundamental problem remains, however. Since all women are fatigued, but only 50% are anemic, we are still left with a policy issue – what good does it do for women to recognize fatigue as a cue for anemia? Perhaps a more dynamic trigger like an increase in fatigue would have been better, but the authors are left with the information they collected. I am still puzzled why the first part of this work was done. UNICEF already reported that 80% of women in the region did not recognize “tiredness” or “weakness” as a symptom of anemia [ref. 26]. In my mind, those two words are equivalent to the word used in this study “fatigue”. Wouldn’t a woman who recognized that she had anemia seek iron folate supplements? If that is true, then this work seeks to confirm the earlier UNICEF finding and to study in more thoroughness the social/cultural underpinnings for this lack of insight. If that is not true, then the problem is not that they do not recognize that fatigue is a cue for anemia, but that they fail to see that anemia should prompt treatment. The rest of your analysis would seem to refute the latter, so we are left with the question of why the first question was asked in this work when the results were already published. There is one other point in the study design that needs to be addressed (briefly). What is the literacy rate in the region? By enrolling only those subjects who could sign written informed consent, were you perhaps excluding women who might be even more likely to have anemia symptoms of fatigue? And one final question. Were the answers from the 14.3% of women diagnosed with anemia any different from those who were not so diagnosed? There are a few typographical errors, but they will probably be recognized by the authors as they rewrite their manuscript one final time. As they address my final issues, I would strongly encourage the authors to see if they can edit it even more tightly. This manuscript is almost ready to be accepted. I would be happy to review any revisions the authors may wish to submit.
--	---

VERSION 2 – AUTHOR RESPONSE

Responses to Comments Provided by Reviewer 1 (reviewer comments are presented in bullets):

- This manuscript is definitely improved, although it could benefit from tight editing to reduce its length. Much of the confusion has been eliminated about recommendations for women to follow should they make the association between fatigue and anemia.

Thank you for your thorough second review of our manuscript. We are happy to know that you found the manuscript to be much stronger, and we are excited for your third read. As suggested, we’ve tightened up some of the writing and cut down the length.

- A fundamental problem remains, however. Since all women are fatigued, but only 50% are anemic, we are still left with a policy issue – what good does it do for women to recognize fatigue as a cue for anemia? Perhaps a more dynamic trigger like an increase in fatigue would have been better, but the authors are left with the information they collected.

This is an important point. We have now added this note in our limitations section (Discussion, p.23). However, we would like note that preliminary findings from later work produced by this project shows that the prevalence of anemia is much higher than reported in previous literature (approx. 2/3rds of women are anemic and most do not know). Additionally, while our qualitative work indicates that most

women are fatigued, we do not yet understand the extent to which this is confounded with anemia—it may be that most women are fatigued because they are anemic. Nevertheless, we agree that change in fatigue would have been a better indicator and our current manuscript now reflects this in the limitations section.

- I am still puzzled why the first part of this work was done. UNICEF already reported that 80% of women in the region did not recognize “tiredness” or “weakness” as a symptom of anemia [ref. 26]. In my mind, those two words are equivalent to the word used in this study “fatigue”. Wouldn’t a woman who recognized that she had anemia seek iron folate supplements? If that is true, then this work seeks to confirm the earlier UNICEF finding and to study in more thoroughness the social/cultural underpinnings for this lack of insight. If that is not true, then the problem is not that they do not recognize that fatigue is a cue for anemia, but that they fail to see that anemia should prompt treatment. The rest of your analysis would seem to refute the latter, so we are left with the question of why the first question was asked in this work when the results were already published.

This is an important consideration, and it is also something about which our team deliberated a lot. We agree that “tiredness” and “weakness” are equivalent to “fatigue”. However, there are a number of reasons why asking about it still makes sense. First, please note the small (albeit important) difference in our study versus the UNICEF study. The way in which UNICEF asks about tiredness is as an assessment of knowledge related to anemia—they assess the extent to which women know that tiredness is a symptom of anemia. Our current study is slightly different, as we attempt to understand if women perceive fatigue as a symptom at all. We find this question important to ask, as a woman who knows fatigue as a symptom of anemia still may not seek medical treatment for it when she experiences it herself; a woman might think “Yes, I know tiredness/fatigue is a symptom of anemia, but that is not what I am experiencing right now, this fatigue is because of the hard work I’ve been doing”. In this way, it is still perceptions of fatigue, and not anemia, that does not prompt treatment. For this reason, we have decided to include the first part of our study (the quantitative components), as we believe capturing the conceptual dissociation between fatigue and seeking healthcare is as necessary as finding gaps in symptom knowledge. Apart from these arguments, the reviewer likely agrees with the premise that just because one study has published a set of findings does not obviate the need to conduct another study asking exactly the same question, simply because that finding is already known. After all, apart from the scientific need for replication, each study also has its own set of limitations and factors that limit its generalizability, and other studies, conducted by other researchers in different contexts reaching the same conclusion is still good for science.

- There is one other point in the study design that needs to be addressed (briefly). What is the literacy rate in the region? By enrolling only those subjects who could sign written informed consent, were you perhaps excluding women who might be even more likely to have anemia symptoms of fatigue? This is an issue we considered during the design of our consent procedure. According to the Indian Government census, 73% of the total population, and 64% of women, in Odisha are literate (now noted in the Discussion section, p 22-23). The data collection team in Odisha did not report any instances of women who were unable to sign the consent due to literacy. However, this still does not tell us if we somehow excluded women (from the denominator, so to speak) who were not literate. We could have made a provision to include women’s thumbprint in lieu of their signatures (as is often done in other settings), but this was not a recommendation from our local IRB. We feel a bit reassured by the fact that the content of the consent form was verbally read aloud. But this is a limitation, and we have included text in our limitation section, stating that we may have undercounted women whose literacy levels were very low.

- And one final question. Were the answers from the 14.3% of women diagnosed with anemia any different from those who were not so diagnosed?

Thank you for this question. While this question interests us greatly, we do not have the adequate power necessary to answer such a question validly—our total sample was made up of 90 participants, only 30 of whom were women of reproductive age. Thus, we are unable to conduct any within-group analyses, such as this, as it would convey incorrect information if we were to generalize such a small

sample to the population.

• There are a few typographical errors, but they will probably be recognized by the authors as they rewrite their manuscript one final time. As they address my final issues, I would strongly encourage the authors to see if they can edit it even more tightly. This manuscript is almost ready to be accepted. I would be happy to review any revisions the authors may wish to submit.

We appreciate the thoughtful comments in this second review. We are pleased to know that you find this version stronger and almost ready for publication. We have read through the manuscript during our edits and have tightened the paper up where appropriate.

Thank you again to the editors at BMJ Open and the reviewers assigned to this paper. We appreciate your time and careful consideration. We are eager for your third review of our manuscript and await your response.

Sincerely,
Authors.

VERSION 3 – REVIEW

REVIEWER	Anita L. Nelson Western University of Health Sciences, USA
REVIEW RETURNED	17-Aug-2020

GENERAL COMMENTS	This paper is still filled with unnecessarily redundancy and should be edited down by a factor of 2, but the authors were not able to accomplish that in 2 separate iterations, so I capitulate: 2 small edits. Page 2 you need a hyphen between anemia and related. Page 12 you need to fill in the name of the University IRB that approved the project. I assume it was not “XXX”. Congratulations!
---

VERSION 3 – AUTHOR RESPONSE

Dear BMJ Open Editorial Team and Reviewers,

Thank you much for your review the edits to our manuscript, titled “Is fatigue a cue to obtain iron supplements in Odisha India? A mixed methods investigation” (bmjopen-2020-037471). We are incredibly proud of the progress that our paper has made and are excited to see the findings disseminated. Our responses to the reviewer’s small comments are as follows:

- **2 small edits. Page 2 you need a hyphen between anemia and related.**
Thank you for noting this, we have made the requested change.
- **Page 12 you need to fill in the name of the University IRB that approved the project. I assume it was not “XXX”.**
Thank you for this request. We have now unblinded the full paper.

We are very excited to hear back from the BMJ editorial office regarding your decision. Thank you again for your time and consideration.